# LEARNING VISUAL PROMPTS FOR GUIDING ATTENTION OF VISION TRANSFORMERS

## ABSTRACT

The attention mechanism in vision transformers (ViTs) plays a key role in visual processing by leveraging contextual information. In this work, we explore the possibility of guiding the attention mechanism to focus more on a region of interest by introducing visual cues to the input while avoiding intervention on the internals of the ViT. This enables interaction with the ViT to prompt the model to capture more information from that area during inference. A recent discovery shows that placing a red circle on the input to the CLIP vision encoder causes the model to extract more information from that marked region. However, we find that this emergent property is unique to CLIP and does not apply to other vision encoders such as DINOv2. Thus, we explore identifying visual patterns that influence the attention of different ViTs through optimisation, instead of resorting to prior knowledge and guesswork. We learn visual cues that added to any input image would redirect the attention of the pre-trained ViT to its location. These learned prompts are optimised in a self-supervised manner, without requiring annotations, and fine-tuning of the vision transformer. [1]

## 1 INTRODUCTION

The attention mechanism Vaswani et al. (2017) in vision transformers leverages contextually relevant visual information from the entire input image to form embeddings for each token. This process dynamically adjusts the representation of each token, allowing the model to focus selectively on relevant areas, and integrates this context into the embeddings. Such context-aware embeddings lead to enhanced recognition of complex visual data Dosovitskiy et al. (2021). In our study, we explore whether it is possible to direct the attention mechanism toward a specific input region by introducing and constructing a marker for pre-trained vision transformer models. We pursue this investigation to better *understand what kind of visual information directs the attention mechanism* of various pre-trained vision transformers and to explore its potential for visual prompting. Visual prompting Gu et al. (2023a) refers to approaches that embed visual cues into input images with the goal of adapting vision foundation models for new tasks.

Adaptation to new tasks typically requires either pre-training the entire foundation model or fine-tuning it, both of which are computationally expensive compared to prompting. Recent works introduce manual prompting techniques, such as placing coloured circles Shtedritski et al. (2023) on images or blurring regions Yang et al. (2024) around the prompted location, demonstrating their effectiveness in adapting the CLIP vision encoder for fine-grained recognition tasks. However, using these manually engineered prompts relies on prior knowledge of biases or emergent properties formed during training. Interestingly, it has been reported that CLIP's training data already contains similar markers and blur effects Shtedritski et al. (2023); Yang et al. (2024), with the marked positions often the focused content of the image, according to the corresponding caption, which may explain the model's responsiveness to these prompts.

In this work, we observe that this emergent property does not translate to other ViTs trained using different learning paradigms. Notably, DINOv2 Oquab et al. (2023), a purely visual self-supervised model, does not respond to red circles. *Identifying visual prompts for a self-supervised encoder like DINOv2 is particularly important because it extracts significantly richer features from images*

---

[1]The code will be publically available.

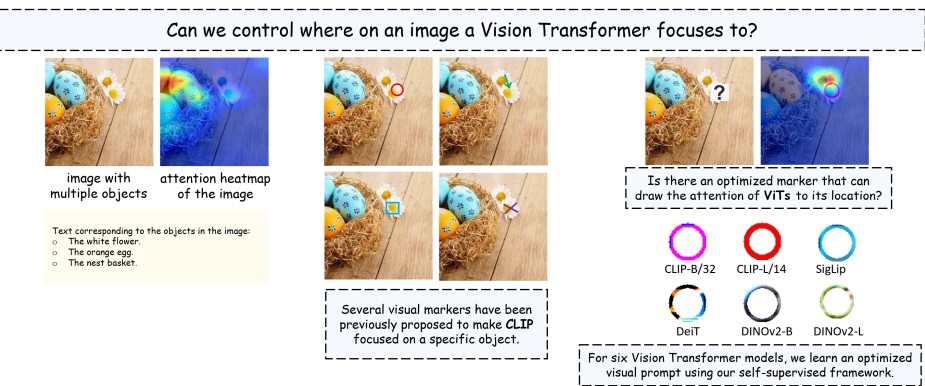

Figure 1: **Learned Prompts for CLIPs** Radford et al. (2021), **SigLIP** Zhai et al. (2023), **DeiT** Touvron et al. (2021), **DINOv2** Oquab et al. (2023). In our self-supervised framework, we learn a prompt to draw the attention of ViT to a specific point where the prompt is applied. The prompt is optimized for each vision encoder model specifically to generalize across different images.

*and is increasingly prominent in the new generation of vision-language models Tong et al. (2024); Kar et al. (2024) due to CLIP's visual limitations* Tong et al. (2024). We analyse attention across layers and discover that adding a red circle to the CLIP vision encoder increases the attention scores in the final layers to the red circle region. Based on this finding, we reverse-engineer the process into an optimization problem Therefore, we propose *learning* prompts to guide the attention of ViTs rather than manually designing them based on intuition. We engineer a visual prompt that directs the attention of a specific pre-trained (and frozen) vision transformer (ViT Dosovitskiy et al. (2021)) through a self-supervised approach. This eliminates the need for prior knowledge about dataset biases, allowing prompts to be generated for a desired ViT. To create a visual pattern that transfers across images for a specific pre-trained ViT, we draw inspiration from universal adversarial patches Luo et al. (2024), leverage a convolutional network prior Ulyanov et al. (2018), and apply the generative adversarial perturbations paradigm Poursaeed et al. (2018). This self-supervised optimization approach generates a visual prompt that can be applied to any location in any image and guides the model's attention during inference without intervention on the hidden layers. The optimised prompts shed insight into what input patterns guide the attention of deep layers of different ViTs, for instance, we find out that the optimal visual pattern for a circular prompt is the "red colour". Moreover, we confirm that directing attention to a region indeed leads to extracting more information from that region.

## 2 RELATED WORK

**Visual Prompting** Prompting has been extensively studied in the NLP community Liu et al. (2023). Recently, researchers have begun exploring the benefits of prompting in image recognition as well. This involves adding a learnable modification to images to guide models towards specific predictions Zhou et al. (2022b); Jia et al. (2022); Bahng et al. (2022); Shen et al. (2024); Gu et al. (2023a). These prompt-tuning methods Jia et al. (2022); Bahng et al. (2022); Shen et al. (2024); Zhou et al. (2022b) optimize a visual prompt that is appended and *fixed* to an input image. For instance, this could involve appending optimizable pixel regions around the image. By doing so, these methods enable the pre-trained model to adapt to new tasks without the need for retraining. This modification is typically applied universally across different images. Follow up approaches have been proposed to improve these visual prompts Zhou et al. (2022a); Zang et al. (2022); Khattak et al. (2023). More recently, researchers have shown that manually crafted prompts, like a red circle, can effectively guide the attention of models like CLIP when trained on datasets containing similar markers Shtedritski et al. (2023). Moreover, other biases, such as blurring, can be used for prompting Yang et al. (2024). However, such attention-guiding prompts only work effectively on certain models. In this study, we propose learning prompts (without fine-tuning the model) to guide the attention of various models rather than manually designing them.

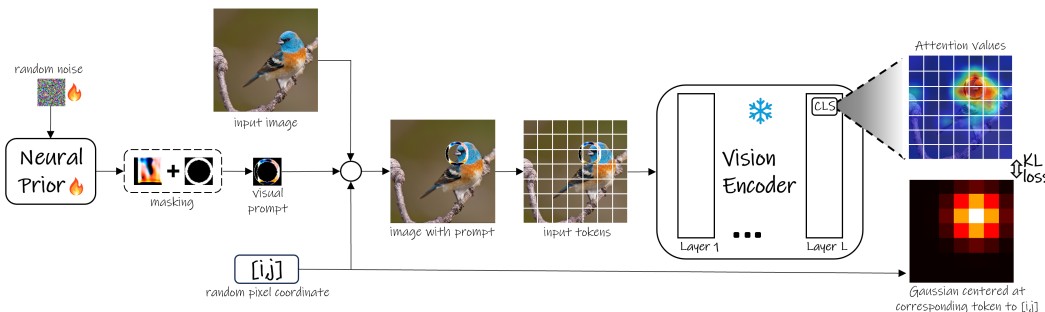

Figure 2: **Overview of our Self-supervised Prompt Optimization Framework:** For a given image random patch position, the patch (initially random noise) is processed through an auto-encoder Neural Prior. A mask is applied to partially cover the image, preserving underlying information. After passing through the frozen Vision Encoder, attention weights are extracted. The target attention values are computed using a Gaussian distribution centered on the target location. During training, the model learns a patch that minimizes the KL divergence loss between the CLS token's attention values and the target distribution.

**Adversarial Patch** The seminal work Papernot et al. (2016) demonstrates that adversarial examples can be generated by altering just a few pixels in the input. This white-box attack Karmon et al. (2018); Liu et al. (2019); Wang et al. (2021); Luo et al. (2021) method can deceive models by targeting a very small area. Additionally, certain studies Brown et al. (2017); Liu et al. (2020) have successfully developed universal, transferable, and targeted adversarial patches. These patches are typically placed on the primary object within images. A related study Gu et al. (2022) found that model attention can be drawn towards adversarial patches designed to mislead classification results. Our visual prompt can similarly be viewed as an adversarial patch, aiming to divert model attention away from its original focus, but for adapting model behavior towards useful tasks.

**Transferability and Universality of Adversarial Perturbation** The transferability of adversarial perturbation describes the phenomenon that perturbations crafted for one model can deceive another, even with a different architecture Gu et al. (2023b). Different from the transferability, the universality of adversarial perturbation is a property that the perturbation is still effective when it is added to different input instances Chaubey et al. (2020). To improve transferability, we use generative adversarial perturbations Poursaeed et al. (2018). A simple way to increase the universality is to include various input images when optimizing an adversarial perturbation Moosavi-Dezfooli et al. (2017), which we also leverage in this work. The transferability and universality of adversarial perturbation have also been studied Brown et al. (2017); Liu et al. (2020); Xiao et al. (2021). These intriguing properties increase the threats in real-world applications. However, how to leverage the two properties for good has not been explored in the community.

## 3 METHOD

Our goal is to learn a patch that, when applied to any part of an image, attracts more attention than the initial underlying pixels of the image from the vision encoder, manipulating the image's final representation. We propose a self-supervised method to learn this patch using a frozen vision-encoder and a collection of unlabeled images. We detail how the patch's RGB color space is learned via back-propagation of a self-supervised loss based on its location and attention values received.

**Input with Prompt.** We aim to discover a visual prompt (patch) that attracts the attention of a transformer-based vision encoder toward a specific location. To achieve this, we work on the RGB image input denoted as $I \in \mathbb{R}^{n \times n \times 3}$ where $n$ is the number of pixels in one dimension of the image and 3 indicates the three-dimensional color space. The prompt, denoted as $\mathcal{P}$, is also in a similar RGB color space and of size $m \times m$ (the impact of prompt size on performance is investigated in section 4.1). The designated coordinates $[i, j]$ within the image $0 \leq i, j < n$ is then the exact pixel location where the visual prompt will be centrally inserted on. By selecting coordinates only with

criteria of $0 < i \pm \frac{m}{2}, j \pm \frac{m}{2} < n$, we ensure that the patch falls within the valid range, meaning that no part of the patch extends beyond the boundaries of the image after it is inserted.

The key point here is that by *inserting* we do not simply mean adding the values of the prompt pixel with those of the image pixel. Rather, we aim to identify a general, universal patch that remains effective when applied to different images and locations, regardless of the underlying pixel values. To achieve this, we *replace* the values of the corresponding image pixels with those of the patch pixels $I[i - \frac{m}{2} : i + \frac{m}{2}, j - \frac{m}{2} : j + \frac{m}{2}] = \mathcal{P}[:,:]$, a process we refer to as *insertion* of the prompt into the image. To ensure the transferability of the visual prompt, the placement of the patch is randomized across various locations within the image during the training. During the training phase, for an input image, we select $k$ random locations denoted by coordinates $[i, j]$ within the validity range criteria mentioned. The patch is then inserted on each coordinate one at a time. This process is repeated for all images $\mathcal{I}$ in the dataset, resulting in a set of modified input images $I_{\mathcal{P}}$ for all samples. These modified images, with patches inserted at random locations, are then fed into the vision encoder.

**Transformer-based Vision Encoder.** Within a transformer block with $L$ layers and $H$ attention heads, we denote the attention values as $A_l^i \in \mathbb{R}^{t \times t}$, where $A$ is the attention values of the $i^{th}$ head of the layer $l$ for an input image with $t$ tokens. The output of the transformer encoder is typically a contextualized representation of the image tokens, where each token has been influenced by its surrounding tokens through the attention mechanism. This representation, often obtained from a special token (e.g., the CLS token), can then be utilized for downstream vision tasks such as object recognition or image classification. Our method leverages the attention values of the CLS token in the last layer, averaged over the heads of that layer: $\bar{A}_L[CLS, *] = \sum_{i=1}^{H} A_L^i[CLS, *]$.

**Target Gaussian Map.** When the prompt is applied to the input image on the pixel position $[i, j]$, it overlays on one or more tiles with the center $[x, y]$, refer to Fig. 2. Assuming that the model divides an $n \times n$ pixel image to $t \times t$ image tokens with each tile having the pixel size $n_t$ ($n/t = n_t$), the corresponding patch center $[x, y]$ can be derived from its pixel position on the image $[i, j]$ as: $x, y = (i/n_t, j/n_t)$.

To construct the target Gaussian map for the patch over the $t \times t$ token space, a 2D Gaussian distribution is employed. The Gaussian map $\mathcal{G}(x, y)$ at location $[x, y]$ is represented by the probability density function of a Gaussian distribution $N(\mu, \sigma^2)$, where $\mu = (x, y)$ denotes the mean (center) of the distribution, and $\sigma$ is calculated from the Full Width at Half Maximum (FWHM), which in our experiments is set to the patch size in token space $m/n_t$: $\sigma = (m/n_t)/(2\sqrt{2\ln(2)})$.

**Neural Prior.** To learn the optimized prompt, we aim to avoid the expensive need to fine-tune the large-scale vision encoder by employing it solely in a frozen state. In this way, the only parameters being updated through the training process are the pixel values of the prompt. However, a potential barrier we may encounter in efficiently achieving the optimal patch is the limited learnable variables within the pixel space Ulyanov et al. (2018); Khakzar et al. (2022). Thus we parameterize the patch input space with a neural network as in Ulyanov et al. (2018). Such parameterization is also shown to improve the transferability of optimized perturbations Poursaeed et al. (2018). we use a neural prior $f$ starting from randomly initialized weights, that receives a random prior input $\eta \in m \times m$ , and outputs an initial prompt $\mathcal{P}_{prior} = f(\eta)$ which is then masked by a predefined shape mask $\mathcal{P} = \mathcal{P}_{prior} * \mathcal{P}_{mask}$ and finally inserted centrally on the $[i, j]$ pixel of the input image $I$ to form the input to the vision encoder: $ViT(I_{\mathcal{P}})$. Our neural prior $f$ employs a CNN-based architecture, in particular a U-Net with three layers of downsampling and three upsampling with Sigmoid activation to ensure the values stay between $[0, 1]$. Leveraging shared spatial patterns among pixels in the patch enables effective optimization during training which facilitates efficient visual prompt learning Krull et al. (2019).

**Objective Function.** Assuming that an image with a patch applied on it $I_{\mathcal{P}} = Insert(I, \mathcal{P}, [i, j])$ is the input to the $ViT$, we define mean of last-layer attention values over attention heads of the vision encoder model as $\bar{A}_l = Attention(ViT(I_{\mathcal{P}}))$. Our objective is to train the deep neural prior to output a prompt $\mathcal{P}$ such that it enhances the attention $\bar{A}_l$ at corresponding token at $x, y = (i/n_t, j/n_t)$ with $n_t \times n_t$ indicating the number of pixels in a token. Having this objective in mind, we calculate the final self-supervised loss as follows:

$$\mathcal{L}(I, [i, j]) = \mathcal{L}_{\text{KL}}(\bar{A}_L[CLS, *], \mathcal{G}(x, y)) \tag{1}$$

---

**Algorithm 1** Learning Prompt with Predefined Shape

---

**Require:** $\mathcal{I}, \eta, n_t$        ▷ Image Collection, Patch Prior Noise, $n_t^2$ = pixel size of a ViT tile
     $n = \text{PIXEL\_SIZE}(I)$                         ▷ Image size: $n \times n$
  1: **for** $I \in \mathcal{I}$ **do**
  2:      $\mathcal{P} \leftarrow f_\theta(\eta)$                     ▷ Generate patch from patch prior
  3:      $[i, j] \leftarrow \text{RANDOM}(n)$
  4:      $x, y \leftarrow i/n_t, j/n_t$             ▷ Find corresponding token coordinates
  5:      $I_p \leftarrow \text{INSERT}(I, P, [i, j])$
  6:      $\bar{A}_{CLS} \leftarrow \text{ATTENTION}(\text{ViT}(I_p))$    ▷ Get the averaged attention values for CLS over all heads
  7:      $l \leftarrow \mathcal{L}_{KL}(\bar{A}_{CLS}, \text{GAUSSMAP}(x, y))$
  8:      $\theta \leftarrow \text{ADAMW}(\theta, l)$       ▷ Update the model parameters using AdamW optimizer
  9: **end for**

---

where $\mathcal{L}_{\text{KL}}$ is the KL-Divergence loss, Throughout the training process, the encoder remains frozen, and only the weights of the neural prior undergo updates as a result of the back-propagating of the defined self-supervised loss.

## 4 EXPERIMENTS

In the following, we first present the learned prompts for CLIP-B/32 and CLIP-L/14, discussing several design considerations and their impacts. We then evaluate the effectiveness of the prompts in the context of Naming Keypoints task. We take it a step further by learning prompts for SigLIP Zhai et al. (2023), DeiT Touvron et al. (2021), and DINOv2 Oquab et al. (2023), each of which either features a slightly different encoder architecture, or has a different training objective function, or varies in pretraining datasets. Finally, we compare how the prompts impact their attention through layers.

**Implementation Details and Settings.** All the vision encoders used are pretrained models available from publicly accessible libraries such as Hugging Face Transformers Wolf et al. (2019) and PyTorch Hub Paszke et al. (2017). Unless specified, we use a learning rate of 1e-3 and batch size of 32 and train our framework for 10 epochs. All experiments were conducted on a machine equipped with a single NVIDIA A100 GPU with 80GB of memory.

**Datasets and Evaluation Metric.** *Training:* Our framework is trained using a subset of ImageNet, specifically 10 categories from Deng et al. (2009); Howard (2019), which provides over 13k diverse samples, enhancing the generalizability of our self-supervised method that does not rely on labeled classes, avoiding the resource demands of using the full ImageNet dataset. *Evaluation:* We evaluate our learned prompt on the CUB-200-2011 test set Wah et al. (2011), containing 5794 bird images with 15 annotated body-part keypoints, by positioning the prompt on these keypoints and measuring the accuracy based on the number of correctly identified body parts. To further evaluate our learned prompt, we test its performance also on the RefCOCO Kazemzadeh et al. (2014), RefCOCO+Kazemzadeh et al. (2014), and RefCOCOgMao et al. (2016) datasets which consist of images, annotated with bounding boxes around objects in it, each of which is paired with expressions. More dataset details are in the appendix.

**Models and Baselines.** Specifically, CLIP-B/32 and CLIP-L/14 are employed for their robust zero-shot learning capabilities, leveraging large-scale vision-language pretraining. SigLIP integrates image and language understanding in a synergistic manner. DeiT and DINOv2 are utilized for their state-of-the-art performance in vision transformer architectures, with DeiT focusing on efficient training and DINOv2 providing self-supervised learning benefits. The baseline for comparison involves using the cropped region over the object bounding box in the RefCOCO dataset family. This approach isolates the relevant object from the surrounding context, allowing for a focused evaluation of keypoint identification accuracy. Additionally, random location baselines are employed to assess the robustness of our method against random visual prompts.

### 4.1 GUIDING ATTENTION TO NAME KEYPOINTS

Table 1: **Vanilla Patches and Scale Constraints.** Accuracies on CUB for CLIP-B/32 and CLIP-L/14 while the patch dimensions are set according to their token sizes.

| Dim | CLIP-B | Dim | CLIP-L |
|-----|--------|-----|--------|
| 32x32 | **10.97** | 14x14 | 12.86 |
| 48x48 | 9.85 | 28x28 | 16.04 |
| 64x64 | 9.71 | 42x42 | **17.41** |
| 80x80 | 7.35 | 56x56 | 13.68 |

We begin with a basic prompt, the **vanilla patch**, a simple filled square without shape masking. We train patches of different sizes for the CLIP-B/32 and CLIP-L/14 models, matching the token dimensions of each. Since CLIP models divide images into tokens (base model: $32 \times 32$, large model: $14 \times 14$), our one-token-sized prompt has varying pixel dimensions accordingly.

Table 1 reports the accuracy of keypoint naming on the CUB dataset. We select patch sizes proportional to the tokens: 1, 2, 3, and 4 times larger for CLIP-L/14. For CLIP-B/32, due to the substantial token pixel size, we opt for prompts that are 1, 1.5, 2, and 2.5 times larger to avoid excessive pixel coverage by the patch. The visualizations of learned prompts are available in the appendix.

The results indicate that for CLIP-B/32, the prompt size equal to 1 token yields the best performance, whereas for CLIP-L/14, the 3-times larger prompt achieves the highest accuracy. Looking into this deeper, this suggests that optimal accuracy is achieved when the prompt covers approximately the same area size on the image ($32 \times 32$ and $42 \times 42$) for both models. This may be because a too-large patch, while better at manipulating the image, also covers more of it, leading to information loss and lower performance.

**Beyond vanilla patch towards using shape masks:** Initially, we used a filled square for the prompt. Previous research suggests that the geometric shape of the patch, particularly a circle or a square, impacts its effectiveness Shtedritski et al. (2023); Bahng et al. (2022), with both square and circle shapes demonstrating strong performance. To investigate the influence of prompt shape on performance in naming keypoints task for CLIP models, we employ a square frame (two concentric squares) and a ring (two concentric circles) as predefined shape masks. We define a thickness factor as $\lambda = \frac{inner\ Diameter}{outer\ Diameter}$ as depicted in Figure 3 and investigate the performance for

Figure 3: Definiion of $\lambda$ for ring and frame masks.

different thicknesses together with different patch sizes, again proportional to the model's token size.

What we observe from the performances in Table 2 is that the circle-shaped prompt yields higher accuracy compared to the square-shaped prompt in general. For CLIP-B/32, the circle prompt achieves the highest accuracy of 11.51, and the best square prompt yields slightly lower accuracy. For CLIP-L/14, the difference is significant: the circle prompt reaches an accuracy of approximately 30.5, whereas the square prompt only achieves about 20.5. The comparison of patch sizes reveals findings consistent with the vanilla patch results. For CLIP-B/32, the optimal patch size remains the same as the default size for one token. In contrast, for CLIP-L/14, the highest accuracy is obtained with a prompt three times larger than one token of the large clip model, the size covering a similar number of pixels.

The visualizations of the best-performing square frame and ring-shape prompts are available in the appendix. Colors of red and pink are predominant for CLIP-B/32 and CLIP-L/14, respectively. It is noteworthy that the optimal prompt learned for CLIP-L/14 is a pure red circle. Even when we use a square shape prior, a red color emerges. Previously, Shtedritski et al. (2023) tried various manual markers using intuition and identified a red circle as an effective prompt. Interestingly, their manual marker seems to be the optimal marker for guiding the attention of CLIP-L/14. Though for CLIP-B/32 the pure red circle is not optimum. This suggests it may be an emergent property that arises with increasing scale.

**Impact of Neural Priors:** In Figure 5 we look into how the neural prior helps our prompt for an enhanced attention manipulation of CLIP vision encoders. We examine the attention averaged values of tokens in the area overlaid by the prompt before and after its application for over 1000 images from MS COCO while the prompt location is random. By analyzing attention values across layers, we can assess the prompt's impact. Since attention values are relative, we define *Attention Gain* as the ratio of the difference of averaged attention values of tokens influenced by the prompt to their

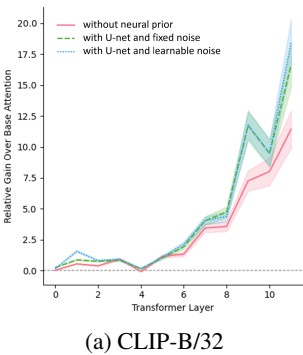

(a) CLIP-B/32

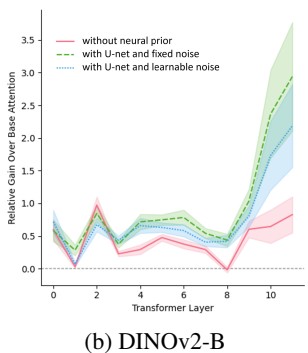

(b) DINOv2-B

Figure 5: **Neural Prior importance:** How the learned prompt influences the attention values across layers with (green and blue) or without (red) neural prior. For both models, attention gain when neural prior is used sensibly increases, emphasizing its beneficial impact.

initial attention values without prompt. Figure 5 illustrates the attention gains for CLIP-B/32 and DINOv2B across their different layers. For both models we observe that U-net neural prior sensibly helps the prompt to learn the optimized pixel values for an increased attention gain. A complete plot for all models is available in the appendix.

## 4.2 INVESTIGATING ATTENTION DYNAMICS IN VARIOUS VISION ENCODERS

Various vision encoder models are trained on diverse datasets, leading to behaviors that may differ from CLIP models. More importantly, these vision foundations use different training paradigms. CLIP vision encoder is trained with language supervision, while DINOv2 is trained by self-supervision using visual information only. Therefore, during training is no supervision signal for the model to learn that colored markers (such as red circles) signify the importance of the region the marker is placed. Additionally, their architectural details might vary slightly, e.g., SigLIP has no CLS token and instead averages over tokens by an attention pooling mechanism. As a result, the optimal prompt to redirect their attention can also be different. To investigate this, we use a similar self-supervised training approach to learn prompts of SigLIP, DeiT, and DINOv2 as well.

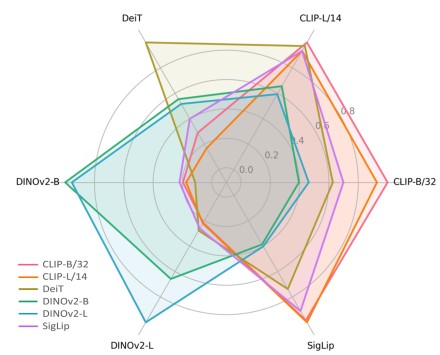

Figure 4: **Applying prompts on different models.** The relative attention gains, normalized by the best-performing prompt of each model.

Table 2: **Impact of prompt's thickness and size:** Accuracies on CUB to correctly identify the bird body parts after applying the learned prompt on keypoints. Varying thickness ratios of frame and ring masks are used over different prompt sizes, proportionate to the model's token pixel size.

| Shape | Thickness | CLIP-B/32 prompt size | | | | CLIP-L/14 prompt size | | | |
|---|---|---|---|---|---|---|---|---|---|
| | | 1.0x | 1.5x | 2.0x | 2.5x | 1.0x | 2.0x | 3.0x | 4.0x |
| Frame | 0.5 | 10.58 | 10.03 | 9.14 | 9.31 | 13.06 | 16.36 | 19.24 | 13.47 |
| | 0.6 | 9.77 | 9.83 | 8.66 | 8.16 | 11.88 | 14.86 | 15.77 | 15.22 |
| | 0.7 | 10.05 | 9.57 | 8.36 | 8.44 | 11.88 | 13.99 | 19.61 | 18.03 |
| | 0.8 | **10.67** | 8.97 | 8.20 | 8.61 | 12.66 | 17.15 | 18.87 | 18.77 |
| | 0.9 | 9.72 | 9.92 | 7.75 | 7.52 | 13.36 | 16.27 | **20.47** | 15.83 |
| Ring | 0.5 | 9.92 | 9.473 | 9.10 | 8.74 | 8.59 | 17.03 | 16.87 | 24.23 |
| | 0.6 | 11.27 | 10.846 | 9.10 | 9.44 | 13.23 | 13.15 | 23.81 | 25.42 |
| | 0.7 | 10.89 | 10.945 | 9.27 | 9.11 | 8.98 | 15.90 | 28.93 | **29.25** |
| | 0.8 | 11.47 | 11.091 | 9.86 | 10.66 | 13.51 | 21.15 | 28.78 | 29.03 |
| | 0.9 | 10.70 | **11.24** | 10.34 | 10.79 | 12.89 | 26.25 | 25.58 | 26.93 |

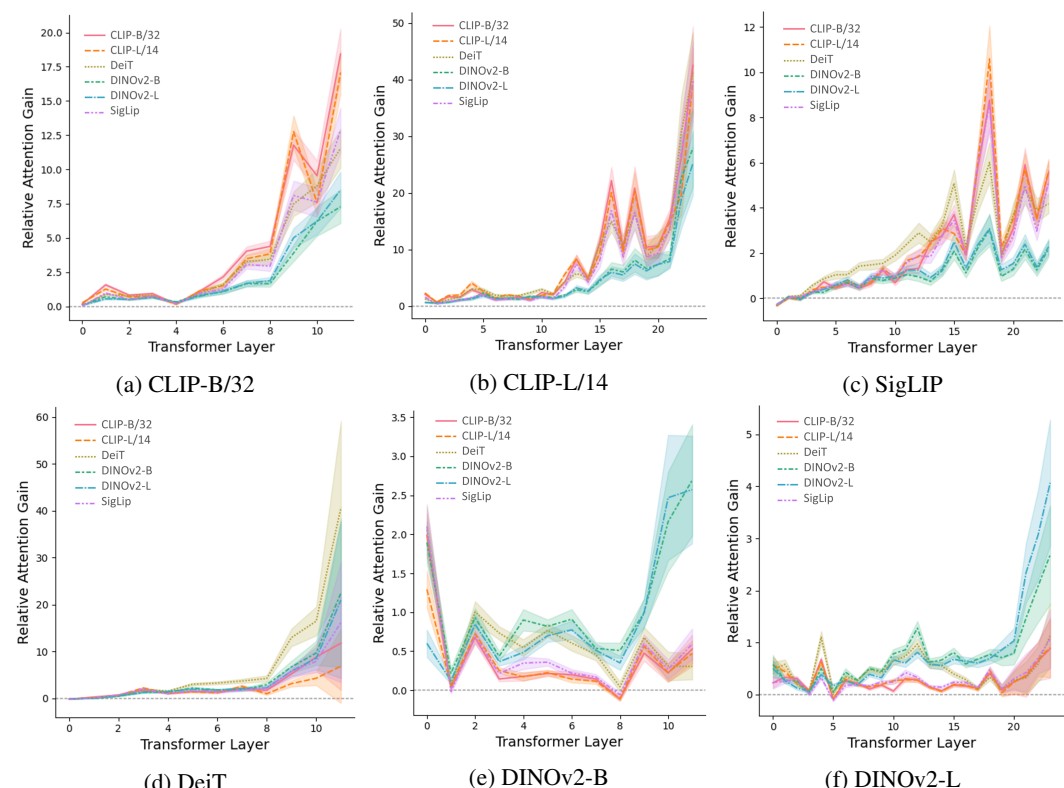

Figure 6: **Attention Gain throughout all layers of different models**. The optimal learned prompt for each model is applied to all other models to evaluate its effectiveness in increasing attention. We observe that each model's own learned prompt consistently achieves the highest attention gain.

Figure 1 shows the visualization of the learned op-
timized prompt for each vision encoder. The different appearance of the visual prompts confirms that there is no single prompt that is universally optimal for all encoders. In Figure 7, we apply the learned prompts on to three different locations on an image and compare the heatmap visualizations of the prompted image with the original, unprompted image. For almost all encoders, the attention heatmap shifts towards the prompt location, demonstrating the effectiveness of our prompt in manipulating the attention of the corresponding visual encoder models.

**Will one model's prompt work for the other?** Figure 6 demonstrates the attention gain of each model when the other model's optimized prompt is applied to the same images. From Figures 6d, 6e, and 6f we can see that the unique prompts for DeiT and DINOv2-B and DINOv2-L draw significantly larger attention to themselves compared to the CLIP-L/14 optimized prompt which is a solid red circle, showing the power of our learning framework. This could confirm the suggestion that the presence of red circles in the training data of CLIP models and SigLIP has made them particularly sensitive to this predefined feature Shtedritski et al. (2023), which is not the case for other vision encoders such as DeiT and DINOv2 models.

Figure 4 compares the attention gains in the last layer of each ViT model across six scenarios, where a single optimized prompt is applied to the image set in each case. The values are normalized by the best-performing prompt for each model. As shown in Figure 4, for nearly all models, their respective optimized prompts yield the highest attention gains. This suggests that a tailored prompt should be learned individually for each model to maximize its performance. More results about the effectiveness of a prompt learned on a combination of models are in the appendix.

To evaluate the performance of learned prompts, we test them on CUB and RefCOCO, which provide annotated image locations for specific body parts (CUB) and objects (RefCOCO). These datasets were not used during training. In particular, evaluation of CLIP-B/32, CLIP-B/32, and

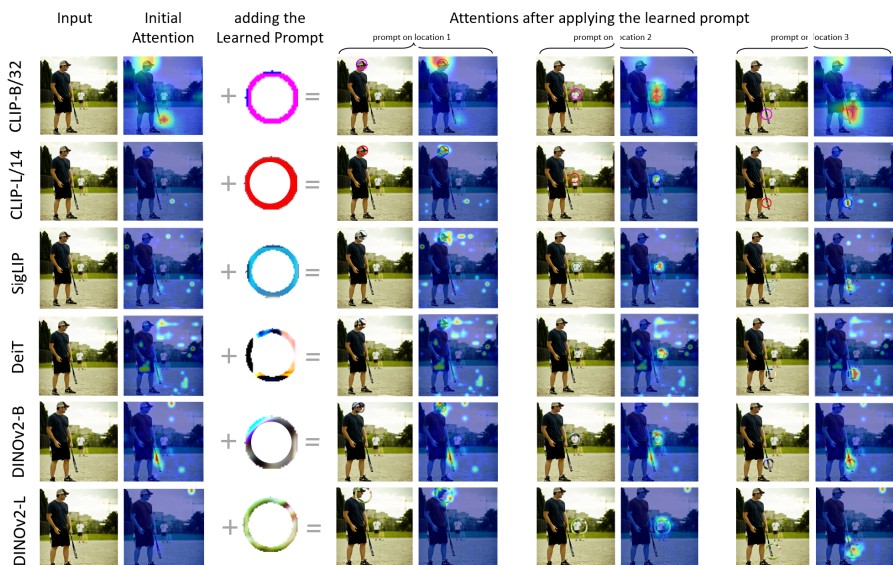

Figure 7: **How Attention Heatmap is changed after Applying the Learned Prompts.** The optimal visual prompts are not the same and each pretrained vision transformers model has its own unique pattern. The prompt is optimized to generalize across images. Comparing attention heatmaps of original (left) and prompted images (right) shows how well prompts focus attention on specific locations.

SigLIP is possible dou to the fact that a text encoder is matched with the embeddings out of their vision encoders. We use the *blurring* strategy as the baseline, which has been identified as the most effective visual marker for annotated data in previous research Yang et al. (2024). The blurring is done by averaging method with a kernel size of 5 over the area around the bounding box for RefCOCO and defining a square similar to the prompt size for CUB, while keeping the area within the bounding box with the original resolution. In Table 3, we can see our learned prompts significantly improve the accuracy on CUB dataset. For RefCoCo, our learned prompt notably exceeds the blurring in CLIP-L/14 and SigLIP. These findings underscore the effectiveness of our approach, especially when the context of the image is important.

## 5 APPLICATION AND FUTURE WORK

**The significance of optimizing prompts for self-supervised models such as DINOv2:** To further evaluate the effectiveness of our prompts, we use the MLLM (Multimodal Large Language Model)

Table 3: **Performance comparison on classification task for different methods across datasets:** While *Blurring* previously showed the highest performance among other location-controllable visual prompts Yang et al. (2024), our learned prompt, specific to the model, demonstrates an exceeded performance for 2 out of 3 models on RefCoCo sets and 3 out of 3 models on CUB.

| Encoder | Prompt | CUB K2N | RefCOCO TestA | RefCOCO TestB | RefCOCO Val | RefCOCO+ TestA | RefCOCO+ TestB | RefCOCO+ Val | RefCOCOg Test | RefCOCOg val |
|---------|--------|---------|-------|-------|-----|-------|-------|-----|------|-----|
| CLIP-B/32 | Blur | 7.4 | **39.6** | **41.1** | **40.0** | **39.5** | **41.2** | **40.0** | 41.3 | 41.1 |
| | ◯ 32×32 | **11.39** | 37.7 | 40.9 | 39.3 | 38.5 | 40.2 | 38.5 | 39.1 | 39.0 |
| CLIP-L/14 | Blur | 11.0 | 42.5 | 41.6 | 41.2 | 43.3 | 41.2 | 41.9 | 42.9 | 43.2 |
| | ◯ 42×42 | **29.7** | **49.5** | **43.9** | **46.7** | **51.3** | **46.2** | **48.1** | **50.4** | **49.0** |
| SigLIP | Blur | 13.1 | 42.9 | 41.3 | 41.8 | 43.6 | 43.0 | 42.8 | 44.2 | 44.5 |
| | ◯ 48×48 | **30.8** | **43.7** | **43.5** | **42.9** | **45.1** | **43.8** | **44.0** | **45.1** | **45.0** |

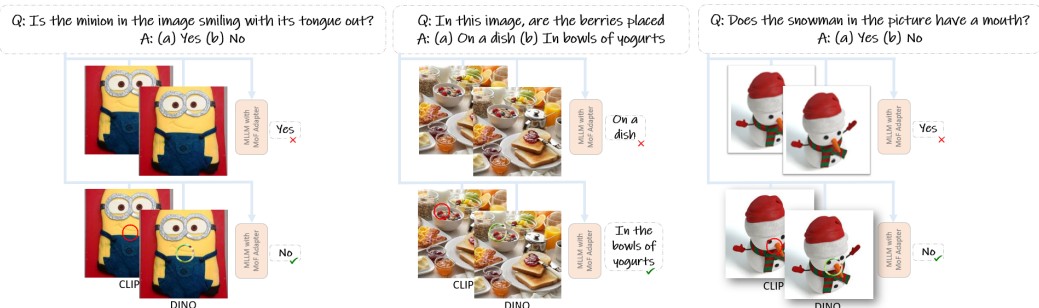

Figure 8: **Prompting applications for the new generation of Vision Language Models:** New vision language models are moving beyond clip vision encoders towards using richer vision encoders such as DINOv2. Therefore it becomes progressively necessary to identify visual prompts for the new generation of models. This figure depicts an example to show the potential of using the optimized prompts in the new generation of vision-language models, in this case, LLaVA+MoF Tong et al. (2024) on examples from the MMVP dataset.

introduced in Tong et al. (2024), which uses an *Interleaved Mixture-of-Features* approach to spatially leverage interleaving CLIP and DINOv2 visual tokens after an adapter. In Figure 8, we compare Tong et al. (2024) model's performance with and without our unique CLIP-L/14 and DINOv2 prompts applied to the image input to see if it improves the question answering responses of their model on proposed MMVP (Multimodal Visual Pattern) Benchmark. The examples demonstrate that our prompts enable the model to generate more accurate answers, which means they create improved embeddings. This indicates that our attention-guiding prompts have promising applications in a variety of vision tasks that can be explored in future works.

**Limitations of our method:** While our method is self-supervised and does not rely on annotated training data, a key limitation is that its application and effectiveness still require annotated datasets, such as CUB, for downstream tasks during inference. An interesting direction for future work would be exploring methods to infer optimal locations for applying our learned prompts without relying on such annotations. Another limitation involves vision-only encoders like DINOv2 and DeiT, which lack zero-shot classification capabilities, preventing us from evaluating our approach on these models, unlike the CLIP family. Moreover, our work is constrained to open-source models, which presents a challenge for learning prompts on proprietary models due to the inaccessibility of their hidden states and attention values. Additionally, in our proposed framework, we only focused on learning prompts within the RGB space and relied on predefined shape masks. Future work could involve incorporating strategies for learning the shapes themselves, further enhancing the model's adaptability and precision.

## 6 CONCLUSION

In this work, we proposed a self-supervised optimization-based visual prompting technique for guiding the attention of vision transformers, thereby avoiding the limitations of manually crafted prompts. Our method demonstrated the ability to guide the attention of various vision transformer models, such as the CLIP family, SigLIP, DeiT, and DINOv2, without requiring prior knowledge of dataset biases and without fine-tuning the models. The transferability of the prompt across different images was accomplished by leveraging a deep network prior. Our experiments confirmed that the learned prompts successfully redirect the attention of not only the CLIP family of models which are trained with language supervision, but also the purely self-supervised models such as DINOv2. As new vision language foundation models are leveraging the self-supervised vision encoders for their superior ability to extract visual features, the proposed optimization-based proves to be helpful for prompting upcoming models.

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

# A  APPENDIX

Table 4: This compares the performance of different methods across various datasets, highlighting that the Crop method excels in RefCOCO tasks, while the Blur method also shows strong results there. In contrast, both methods struggle in the CUB dataset, where contextual information is essential for accurate animal detection.

| Encoder | Prompt | CUB | RefCOCO | | | RefCOCO+ | | | RefCOCOg | |
|---|---|---|---|---|---|---|---|---|---|---|
| | | K2N | TestA | TestB | Val | TestA | TestB | Val | Test | val |
| | Random | 8.2 | 36.9 | 39.5 | 37.8 | 36.9 | 39.7 | 37.9 | 39.5 | 39.1 |
| CLIP-B/32 | Crop | **15.6** | 56.6 | 47.7 | 52.6 | 60.4 | 51.7 | 55.9 | 60.3 | 59.5 |
| | Blur | 7.4 | 39.6 | 41.1 | 40.0 | 39.5 | 41.2 | 40.0 | 41.3 | 41.1 |
| | ⬭$_{32 \times 32}$ | 11.39 | 37.7 | 40.9 | 39.3 | 38.5 | 40.2 | 38.5 | 39.1 | 39.0 |
| CLIP-L/14 | Crop | 18.5 | 58.8 | 47.8 | 53.5 | 63.5 | 51.4 | 57.5 | 61.2 | 60.6 |
| | Blur | 11.0 | 42.5 | 41.6 | 41.2 | 43.3 | 41.2 | 41.9 | 42.9 | 43.2 |
| | ⬭$_{42 \times 42}$ | **29.7** | 49.5 | 43.9 | 46.7 | 51.3 | 46.2 | 48.1 | 50.4 | 49.0 |
| SigLIP | Crop | 19.6 | 62.6 | 51.3 | 57.9 | 66.8 | 56.7 | 62.3 | 69.2 | 68.5 |
| | Blur | 13.1 | 42.9 | 41.3 | 41.8 | 43.6 | 43.0 | 42.8 | 44.2 | 44.5 |
| | ⬭$_{48 \times 48}$ | **30.8** | 43.7 | 43.5 | 42.9 | 45.1 | 43.8 | 44.0 | 45.1 | 45.0 |

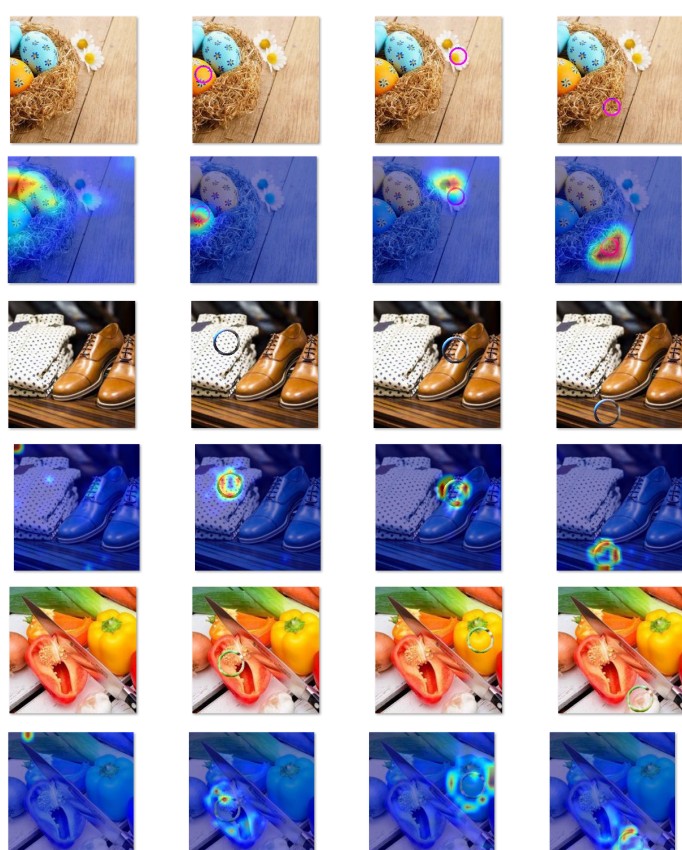

Figure 9: With our learned prompts, the model's attention can be redirected to any location where the prompt is inserted, regardless of the background colors or context.

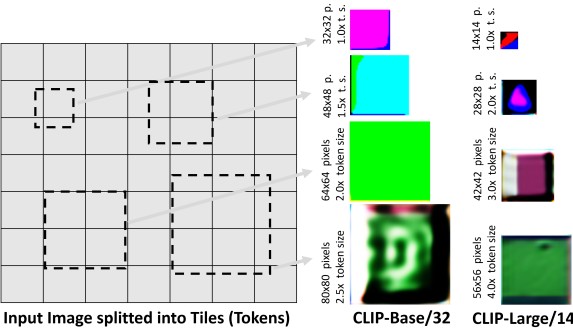

Figure 10: Visualizations of vanilla patches learned for CLIP family models. Their size is proportional to the token size of the model as demonstrated in the gray area.

The prompt has been developed, and the design configurations have been explored. Interest now lies in evaluating its performance compared to baseline methods on different datasets. To this end, the method is trained on the ImageNet dataset and tested on the CUB and RefCOCO datasets. These datasets are new to the prompt and were not seen during the training phase. Additionally, the datasets offer the advantage of annotated image locations with names of specific body parts or objects for CUB and RefCOCO, respectively.

The baselines are defined as follows: *Random*, where areas are randomly selected; *Crop*, where the area around the bounding box is removed for RefCOCO, and a square similar to the prompt size is defined for CUB before cutting out that area; and *Blur*, which is similar to cropping but

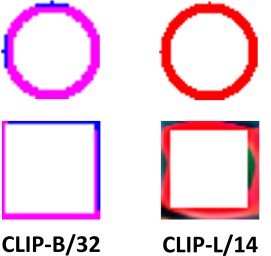

**CLIP-B/32**  **CLIP-L/14**

Figure 11: Visualizations of the prompts with the best performance on CUB dataset from Table 2 in the paper. We see that the red and pink color is predominant for both circle and rectangle shapes.

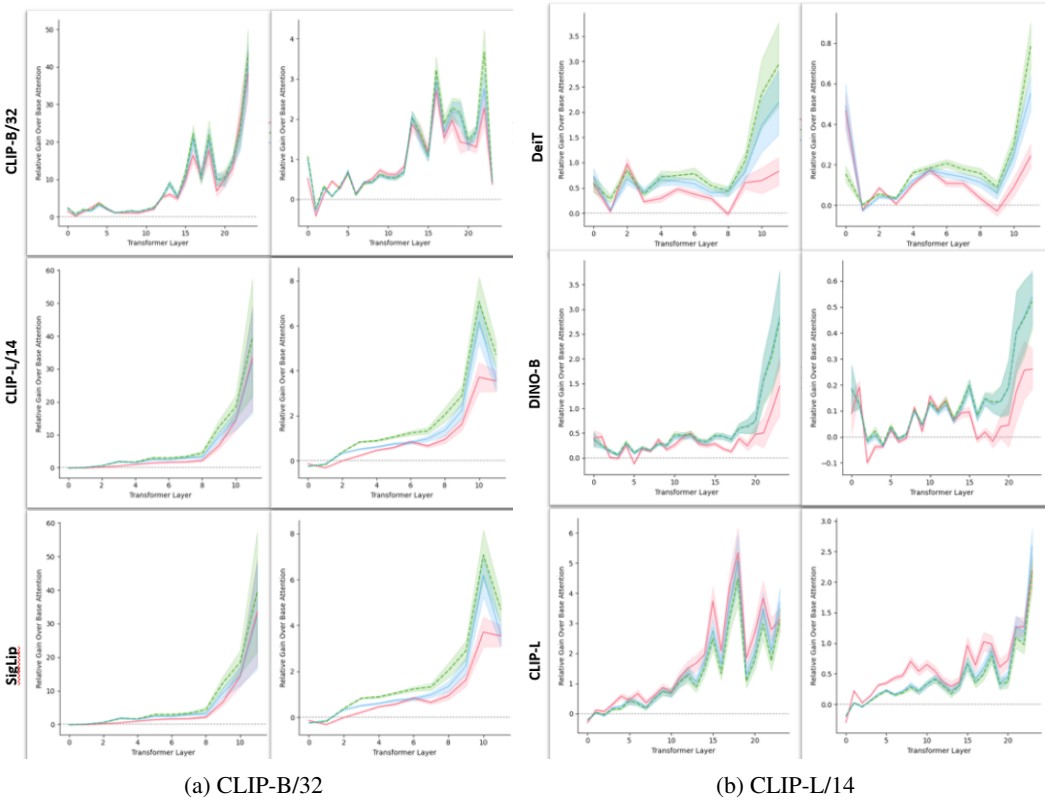

(a) CLIP-B/32  (b) CLIP-L/14

Figure 12: **Attention Gain of all models with and without neural prior.** We applied our learned prompt to random locations on 1000 samples from the MS COCO dataset. We try to show how the neural prior effects the improved performance of prompt in terms of attention gain.

with the outer area blurred using the averaging method with a kernel size of 5. As indicated by the results presented in Table 4, superior performance in RefCOCO tasks is exhibited by the Crop method, though its effectiveness diminishes when applied to the CUB dataset. In contrast, notable improvements in RefCOCO tasks are shown by the Blur method.

It has been observed that RefCOCO object descriptions often focus solely on the target object rather than relying heavily on scene context. This tendency explains the success of both the Blur and Crop methods in this dataset. However, in the CUB dataset, the accurate recognition of animals is challenging for the vision encoders, making it difficult for specific body parts to be detected. Despite this challenge, the learned prompts generally outperform other baselines in the CUB dataset. In RefCOCO tasks, second place is consistently achieved by the method, after the effective Crop method. These findings underscore the effectiveness of the approach, especially when the context of the image is important.

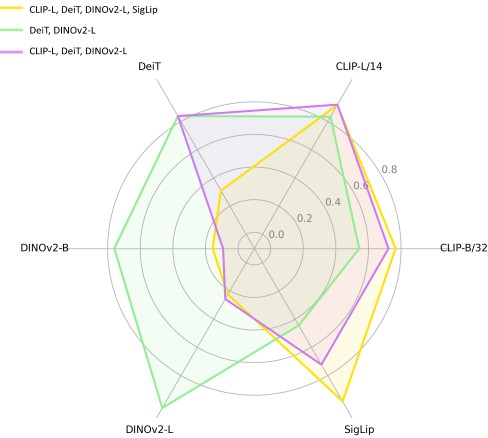

Figure 13: When we trained the optimized prompt for a combination of models, their performance in terms of normalized attention gain is different. Here we can conclude that there is no global prompt that will perform best in all Vision Encoders at the same time.

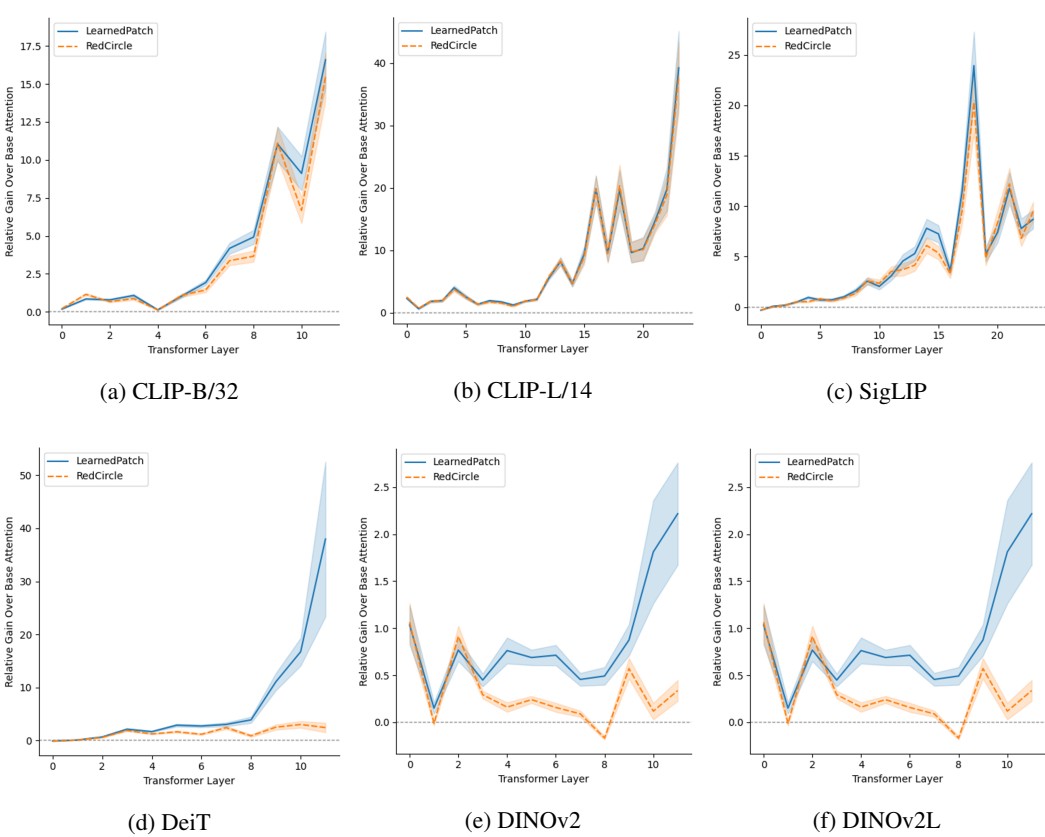

Figure 14: Attention Gain throughout all layers of different models. The optimal learned prompt is compared with the baseline of a simple red circle manual marker. In CLIP family, the red circle as a prompt is effective. However, for more recent models like DeiT and DINO family, this is too simplistic. Our learned prompts exceed the manual marker of a red circle by far.

