# OpenReview forum: "Learning Visual Prompts for Guiding the Attention of Vision Transformers"
_ICLR.cc/2025/Conference — Submitted to ICLR 2025_

### Official Review · Reviewer_udzU · 2024-10-16

**Soundness:** 2
**Presentation:** 2
**Contribution:** 2
**Rating:** 3
**Confidence:** 5

**Summary:**

This paper explored the possibility of guiding the attention mechanism to focus more on a region of interest by introducing visual cues to the input while avoiding intervention on the internals of the ViT. However, I have some concerns as follows:

**Strengths:**

1. The structure of this paper is complete.
2. The content of this paper is easy to follow.

**Weaknesses:**

1. The Introduction is confusing. There is no logical relationship between the three paragraphs of the introduction, and it is unclear what the author is trying to do within the three paragraphs. How do existing methods do it? What problems do they have? What is the difference between the proposed method and the above methods? What is different and innovative about the proposed method?
2. In related work, what is the relationship between the three parts of the introduction? What problems do existing methods have that should be further elaborated?
3. The proposed method is too simple. It feels like a patchwork of existing methods. Compared with existing methods, I don’t know where the core innovation of the proposed method is.
4. The experimental results are unreliable. First, there is a lack of comparison with the most advanced methods. Second, the analysis is insufficient for ablation studies. It is unclear why each of the proposed components can bring performance improvements.

**Questions:**

See the comments below.

---

> ### Author Response · Authors · 2024-12-04
>
> Thank you for your feedback. As outlined in the introduction, existing methods for redirecting the attention of ViTs to specific locations either require fine-tuning or rely on manually placed markers on the input image. The former is computationally expensive, while the latter introduces human bias, making it far from an optimal solution.
>
> .
> We are the first to achieve a visual marker capable of directing the attention of ViTs to specific locations using a self-supervised learning strategy. Unlike previous markers, which are manually engineered, our approach represents a significant step forward. Not only does it demonstrate superiority and innovation by learning the optimal marker, but it is also model-agnostic, making it applicable to all types of transformer-based vision encoders. Moreover, our method operates in a fully self-supervised manner, eliminating the need for costly annotated data. This further emphasizes its strength in terms of transferability across diverse datasets. We would appreciate a deeper explanation of why such a groundbreaking approach might be viewed as less innovative compared to manual, heuristic methods, as this perspective seems to overlook the clear advantages of our contribution to manipulating the attention of vision-transformers.
>
> .
> While the manual markers may appear straightforward, the learning strategy we propose is far from simplistic. It employs a carefully designed self-supervised loss function to bridge two distributions: the Gaussian attention map and the averaged attention values of the CLS token. This is a delicate process, as the values in the pixel space are not directly updated but instead rely on a neural prior (as illustrated in Figure 2 and ablated in Figure 5) and a random noise initialization patch. We would greatly appreciate more detailed feedback on how the reviewer perceives our approach as simple, particularly in comparison to previous methods for location-controlled visual markers. These earlier methods use heuristic techniques to draw markers directly on the input image, which contrasts sharply with the principled and sophisticated framework we present.
>
> .
> Currently, no learning-based method exists for optimizing visual markers in Vision Transformers (ViTs). We are the first to propose such a method. In contrast, all prior approaches heavily rely on manually engineered markers derived from human intuition (see Shtedritski et al., 2023, and Yang et al., 2024). To address this gap, we thoroughly compare our method against the strongest engineered marker to date—the blurring strategy (Yang et al., 2024)—in Table 3. Further, for the three models aligned with text encoders (CLIP-large, CLIP-base, and SigLip), we report accuracy based on the number of correctly identified body parts, demonstrating the clear advantage of our learned markers in Table 3. For the remaining vision encoders (DeiT, DINOv2-Large, and DINOv2-Base), which lack aligned text encoders, we propose to look at he improved attention on the pointed region. We highlight the effectiveness of our learned marker through its measurable influence on attention values. Figure 6 explicitly shows how the attention values across transformer layers improve when our learned prompts are applied. Moreover, we demonstrate that markers optimized for one model (e.g., CLIP-large) are not optimal for another (e.g., DINOv2), underscoring the specificity and adaptability of our method.
>
> .
> In addition to this, Tables 1 and 2 present detailed ablations on marker size and shape (vanilla, frame, and ring) with varying thickness factors. Figure 4 further illustrates the effect of applying learned markers interchangeably between models, revealing their direct impact on attention values across transformer layers. In Figure 5, the significance of the neural prior component is ablated. Importantly, this work is unique and the first to examine how attention values evolve through ViT layers when applying visual markers. Figure 7 provides compelling evidence through attention heatmaps, showing how our learned markers dynamically redirect attention to specific objects in an image. Finally, Figure 8 demonstrates the transformative power of our prompts in Visual Question Answering (VQA) tasks. Without prompts, the language model produces incorrect answers; with our learned prompts, it returns accurate responses.
> We request the reviewer to address the shortcomings in our work. If these extensive experiments, ablations, and evaluations are still considered insufficient, we respectfully request specific, actionable feedback detailing what is missing. Given the breadth and depth of our study, we find it essential to understand how our approach, with its significant innovation of proposing a self-supervised optimization strategy for learning visual markers, could be deemed inadequate compared to heuristic, manually engineered methods. We hope this explanation clarifies our decisions and approach.

---

### Official Review · Reviewer_Hb9h · 2024-10-27

**Soundness:** 2
**Presentation:** 3
**Contribution:** 2
**Rating:** 3
**Confidence:** 4

**Summary:**

This paper proposes a self-supervised optimization approach for designing visual prompts to guide the attention of vision transformers (ViTs) toward specific regions in an image. Instead of manually creating visual markers, the method involves learning visual prompts specific to each vision encoder, enhancing its focus without altering its architecture. Experiments on CLIP, DINOv2, and other ViTs show that these prompts effectively influence attention distribution, improving model performance on tasks like keypoint naming and object recognition.

**Strengths:**

The approach appears to be adaptable, allowing self-supervised learning of visual prompts for various vision encoders without needing labeled data. This flexibility is particularly valuable for adapting vision transformers across tasks and models without costly retraining or fine-tuning.

**Weaknesses:**

There are a few concerns for the algorithmic design: 1. Although the method is intended as a self-supervised approach, its performance is primarily evaluated using annotated datasets like CUB (for keypoint detection) and RefCOCO (for object localization), where specific body parts or objects are pre-labeled. This reliance contrasts with the self-supervised nature of the learning process, potentially limiting practical utility. For instance, if access to labeled datasets is restricted or unavailable in a particular domain, the evaluation and validation of the learned prompts could become challenging, affecting the applicability of the approach in real-world, unlabeled environments. 2. The evaluation is performed on datasets with relatively controlled settings (e.g., images with birds or single annotated objects). In more complex, natural scenes where multiple objects and overlapping contexts are present, the effectiveness of learned prompts could degrade. Without sufficient evaluation on diverse, unlabeled datasets, there’s a risk that the method may not generalize effectively to scenarios with complex visual contexts. This gap highlights a limitation in validating the approach’s robustness and universality outside curated datasets. 3. The learned prompts are customized for individual transformer-based architectures (e.g., CLIP, DINOv2), which may not translate effectively to other model types or even to transformers with slight architectural variations. This specificity can limit cross-model generalization, requiring separate training of prompts for each new architecture. For broader applicability, it would be beneficial to explore if and how prompts could be designed with more universal features or optimized independently of model architecture.

**Questions:**

In light of these weaknesses, I would suggest the authors address the following: 1. How does the method perform on real-world datasets with complex and diverse scenes? The paper’s focus on curated datasets like CUB and RefCOCO raises questions about robustness on uncurated, real-world datasets with more complex scenes. Such environments may demand attention guidance that adapts beyond controlled settings, so further validation in these contexts would be beneficial. 2. Could the approach be generalized to non-transformer vision models? As the study is limited to transformer-based models like CLIP and DINOv2, testing with other architectures (e.g., CNNs) could clarify whether the learned visual prompts are uniquely suited to transformers or if they have broader applicability across different model types. 3. How can we ensure the accuracy of attention guidance? Since the model’s "attention" relies on pre-trained representations, this approach may lack robustness across different test set distributions (e.g., open-set scenarios). The work’s effectiveness hinges on accurate attention to guide model behavior, but if the attention is inconsistent, it could potentially mislead the model, impacting performance.

---

> ### Author Response · Authors · 2024-12-04
>
> Thank you for your thoughtful feedback. We respectfully disagree with the assertion that the use of labeled datasets for evaluation contradicts the self-supervised nature of our training approach. It is important to clarify that using a self-supervised training method does not inherently require a self-supervised evaluation. While self-supervised evaluation can be valuable in scenarios where labeled evaluation data is unavailable, performance metrics derived from labeled, annotated data are often more reliable and interpretable. This is especially true for benchmarking against widely recognized datasets like CUB (for keypoint detection) and RefCOCO (for object localization), which offer standardized comparisons across methods. Furthermore, we do address the scenario of label-scarce environments in our work. As demonstrated in Figure 6, we propose a self-supervised evaluation metric that measures attention gain, providing an alternative means of validating the effectiveness of our method without relying on labeled data. This approach showcases the versatility and practicality of our method in both labeled and unlabeled contexts.
>
>
> We acknowledge the importance of evaluating methods in diverse and challenging settings. In our work, we strive to include a wide range of evaluation scenarios to test the robustness of our approach. Specifically, we assess performance on single-object test datasets like CUB, multi-object annotated datasets such as RefCOCO, and sampled data from the COCO dataset with randomly assigned target locations. These settings collectively represent both controlled and less structured visual contexts. Moreover, the use of the Gaussian target map in our self-supervised training ensures that the model focuses on the area surrounding the prompt's location, regardless of the presence or type of object in the image. This design inherently encourages generalization by making the model robust to variations in object appearance and context. While more evaluations on completely unlabeled, complex natural scenes are valuable future directions, we believe our current setup demonstrates the method's applicability across a range of scenarios, balancing complexity and practicality.
> Prior to this work, other studies have proposed manually crafted prompts, claiming these prompts are universally applicable across all vision transformer (ViT) models. However, our results demonstrate that the effectiveness of such prompts heavily depends on the training procedures (e.g., datasets and objectives) of the tested ViTs, and they fail to generalize effectively to newer models like DINO and DeiT (as shown in Figure 6). This indicates that these so-called "universal" prompts are not as broadly applicable as previously claimed. The primary goal of our work is to learn prompts tailored to each individual model, removing reliance on prior biases stemming from the specific training procedures of these models. Furthermore, as shown in Figure 5, learning a single universal prompt for multiple models leads to performance degradation across a selection of ViTs when compared to individually learned prompts. While such a universal learned prompt still significantly outperforms manually crafted prompts based on human intuition, it highlights the challenge of generalizing prompts across diverse architectures. For each model, we only need to train.
>
>
> Lastly, about the comparison with the CNN architectures mentioned in the questions, we clearly indicate in the title that this work is only proposed because we are presenting a self-attention mechanism that does not exist in the CNN methods! We hope this explanation clarifies our decisions and approach, and we thank you again for your valuable feedback.

---

### Official Review · Reviewer_LF2a · 2024-10-31

**Soundness:** 2
**Presentation:** 2
**Contribution:** 2
**Rating:** 5
**Confidence:** 3

**Summary:**

This paper introduces a self-supervised learning approach for generating visual prompts that guide the attention mechanism of Vision Transformers (ViT). The method utilizes a neural network prior and optimizes the prompt pixels through backpropagation to enhance attention at corresponding positions in the visual encoder. The approach is evaluated on the CUB and RefCOCO datasets, showing improved performance compared to baseline methods.

**Strengths:**

1. The approach is model-agnostic and does not require fine-tuning, making it versatile across different models.
2. The method demonstrates strong generalization capabilities across various models and datasets.
3. The presentation is clear and easy to understand.

**Weaknesses:**

1. The paper lacks sufficient justification for the advantages of the approach compared to directly collecting and fine-tuning with annotated visual prompt data, particularly in terms of performance and efficiency benefits.
2. There are some typos: some citation formats need correction; Table 2 shows inconsistent decimal places across the results; the ordering in Figures 4 and 5 appears to be incorrect.

**Questions:**

1. When applying multiple visual prompts simultaneously, do the attention maps exhibit similar patterns of change as observed with single prompts?
2. How does the performance compare quantitatively to using a simple red circular frame as a visual prompt?
3. Could you consider using attention gain as a training objective to enhance the model's ability to follow visual prompts with LoRA (a efficient training method)? This could provide an interesting baseline comparison.

---

> ### Author Response · Authors · 2024-12-04
>
> Thank you for your insightful feedback. We appreciate the opportunity to address your points and clarify our decisions.
> First, regarding the computational requirements, fine-tuning a ViT foundation model is significantly more computationally demanding than learning a limited pixel-space patch and a lightweight, 3-layer CNN-based neural prior. For this reason, we did not consider an ablation study on fine-tuning to be particularly insightful for inclusion in the paper. As noted in the Introduction, we emphasize the high computational cost of fine-tuning, but since our approach circumvents this need, we did not find it necessary to include additional experiments on this topic in the Results section.
>
>
> We also acknowledge the suggestion to explore multiple markers on a single image as a potential future direction and will consider this as an additional ablation in a subsequent version of the submission. However, it is important to note that this was not the primary goal of our study. Our focus was on redirecting the attention of a ViT to a specific location. The objective function of our method was explicitly designed around this goal, targeting a single attention point rather than multiple markers. In Figures 4, 5, and 6, we compare our approach with the red circle prompt, studying the attention gain across all model layers. As stated earlier in the paper, the red circle has been identified as the optimal prompt for the CLIP-Large model. In Figure 6, we include the red circle alongside all the learned prompts for various models to ensure a comprehensive comparison.
>
>
> While LoRA is indeed a popular PEFT method, it still involves significantly more parameters than our approach, which is limited to a 3-layer neural prior and a token-sized learnable pixel patch. The primary goal of our research is to discover the optimal patch while leveraging frozen ViTs. This optimal patch not only provides computational efficiency but also enhances the interpretability of ViTs by shedding light on their attention mechanisms. Our learned prompt successfully manipulates the model’s attention to a specific location without requiring any structural changes or fine-tuning of the model itself, maintaining its frozen state. We hope this explanation clarifies our decisions and approach, and we thank you again for your valuable feedback.

---

### Official Review · Reviewer_EsHF · 2024-11-03

**Soundness:** 1
**Presentation:** 2
**Contribution:** 1
**Rating:** 3
**Confidence:** 5

**Summary:**

This paper focuses solely on learning prompts through self-supervised learning, which can be added to any input image to redirect the attention of the pre-trained ViT to its location. It achieves this by minimizing the KL divergence between the attention of the CLS token and the target distribution.

**Strengths:**

- The paper is easy to follow and understand.
- This method can be applied to different pre-trained ViT, regardless of their pre-training supervision methods, and does not require annotation or fine-tuning of the ViT.

**Weaknesses:**

- The paper lacks essential details and descriptions, which may lead readers unfamiliar with the preceding work to struggle with understanding what is being tested or implemented. For example, the methodology for testing on the CUB dataset is not clearly outlined, and the significance of "K2N" on line 478, which stands for keypoint to name, is not explained.
- Building on the concept that a "red circle" can direct CLIP's attention to a specific area, as demonstrated in prior work by Shtedritski et al. (2023), the paper's technical contribution is limited, and its effectiveness is also open to debate:
    1. It primarily applies the phenomenon of the "red circle" to other pre-trained Vision Transformers through learnable prompts. However, the experiments involving language-supervised methods lack baselines for the "red circle" and cropping techniques. Moreover, the performance comparison with the "red circle" from the referenced paper [https://arxiv.org/pdf/2304.06712] appears to show a decline rather than an improvement.
    2. In the DINOv2 experiments, the paper shows examples from MMVP for illustration but does not provide quantitative metrics.
    3. The paper combines the phenomenon of the "red circle" with visual prompt learning, which is incremental. Moreover, its effectiveness has not been well demonstrated.
- The application scenarios are limited: the phenomenon of the "red circle" is quite interesting and helps us better understand CLIP. This paper focuses on applying this phenomenon to more pre-trained Vision Transformers. However, the method of using circles necessitates manually annotating areas of interest on images, making it impractical for large datasets. Consequently, effectively applying this phenomenon to downstream tasks is challenging. If possible, it should be stated how the proposed method can be applied to downstream tasks.
- There is a lack of suitable baselines and quantitative comparisons. In the absence of an appropriate baseline, it is challenging to demonstrate that the proposed method improves upon previous approaches. Since the paper’s contribution is to extend this phenomenon to other ViTs, it is essential to conduct controlled ablation experiments on the MMVP-VLM dataset to show that the proposed method is more effective than the simple "red circle" approach.
- Minor: The presentation of the figures needs improvement; for instance, in Figure 7, the text for location 2 3 is obscured.

**Questions:**

Please refer to the Weaknesses section for further details.

---

> ### Author Response · Authors · 2024-12-04
>
> We appreciate your comments and feedback. In the Methodology section, we have clearly and meticulously detailed the components of our proposed framework: the initialization of random noise, the neural prior, the shape mask, the randomized insertion of the learned marker onto images, and the use of a frozen vision encoder as the main model. Furthermore, we explicitly describe our self-supervised KLD loss function, which aligns the attention heatmap with the Gaussian map defined by the random location of the learnable marker. These steps are thoroughly documented. In the Experiments section, we dedicate a full paragraph to outlining the training datasets. For evaluations, the relevant test sets are explicitly stated alongside the respective figures and tables. For instance, in Figures 4, 5, and 6, we clarify that attention gain values were calculated on a random subset of MS COCO, demonstrating the generalizability of our learned prompt trained on ImageNet. Tables presenting accuracy results clearly specify the CUB or RefCOCO family as the evaluation datasets, leveraging their ground truth text annotations. Additionally, the Keypoint-to-Name task is a well-known benchmark, and due to space constraints, we deemed an extended explanation unnecessary.
>
>
> It is deeply surprising and disappointing that the reviewer has overlooked these details and dismissed our work as an incremental contribution. Our framework introduces a **self-supervised learning strategy** to direct the attention of transformer-based vision encoders (e.g., CLIP, DeiT, DINO) to specific image locations—an innovation that extends well beyond prior studies. In the Introduction, we emphasized the contrast with previous works, such as the "red circle" manual markers, by proposing an entirely learnable framework. Furthermore, we included a comparison with image cropping in the appendix. However, it is unfair to remove major contextual parts of an image in the cropping baseline while our marker allows the full image context to be preserved. Comparisons with the “red circle” baseline were extensively evaluated through our self-supervised metric, **attention gain**, as demonstrated in Figures 4, 5, and 6. These results highlight that our learned markers, tailored specifically to each model, outperform generic manual markers. Notably, while prior works were limited to the CLIP family, our contribution extends beyond to include **SigLip, DeiT, and DINOv2**—a significant step forward that should not be trivialized. This work presents a clear advance in self-supervised learning for vision encoders, offering a novel method to optimize attention redirection across diverse transformer-based models. We respectfully urge the reviewer to reconsider the depth and scope of our contribution.
>
>
> Finally, while the LLaVa + Mixture-of-Features model by Tong et al. (2024) requires location information due to the MMVP dataset’s task-specific nature, this dependency is not methodological. We explicitly acknowledge this in the paper’s limitations and future work sections. These points were transparently presented, and any misinterpretation of our approach is unfounded. The experiment on the MMVP dataset (Figure 8) demonstrates the utility of visual markers for next-generation VLMs. It shows that applying a prompt to a specific location enhances language output by redirecting the vision encoder’s attention, enabling the language decoder to produce more accurate answers.
>
> The true comparison, as depicted, lies between the outputs with and without prompts. We acknowledge the reviewer’s suggestion and will include statistics on successful cases in future versions. We argue that manually engineered markers like "red circles," though effective, may succeed by chance, with no evidence of their optimality for directing CLIP’s attention. Previous studies have focused solely on CLIP, with no exploration of other ViT-based models. In contrast, our framework removes biases from manual engineering, leveraging optimization to control attention mechanisms across models like CLIP, DeiT, and DINO.
> Regarding marker placement, during training, markers are inserted randomly, ensuring a self-supervised approach independent of annotated data for both location and text. In testing, evaluations were conducted in supervised and unsupervised settings. The supervised approach requires location and object-specific text for accuracy measurements, as shown on CUB and RefCOCO datasets, which we explicitly acknowledge as a limitation. The unsupervised evaluation, based on attention gain, demonstrates independence from text annotations or object locations, with experiments on MS COCO (Figures 4, 5, and 6) highlighting the flexibility of our approach in the absence of labeled data. We hope this explanation clarifies our decisions and approach, and we thank you again for your valuable feedback.

---

### Author Response · Authors · 2024-12-04

### Overall Comment for ICLR Rebuttal:

We thank the reviewers for their insightful feedback and the opportunity to clarify key aspects of our work.

First, we emphasize that our framework is the first to propose a **self-supervised learning strategy** for optimizing visual markers in Vision Transformers (ViTs). Unlike prior heuristic approaches, such as manually engineered markers ("red circles") or computationally expensive fine-tuning, our method eliminates human biases, operates in a fully self-supervised manner, and is applicable across diverse ViT-based models (e.g., CLIP, DeiT, DINO). This significant innovation is supported by extensive experiments, as outlined in Figures 4–8 and Tables 1–3, and highlights the adaptability of our method to different architectures.

We respectfully clarify that using labeled datasets for evaluation does not contradict the self-supervised nature of our training. Labeled benchmarks (e.g., CUB and RefCOCO) provide standardized and reliable performance comparisons. At the same time, we address label-scarce environments through our self-supervised metric, **attention gain**, as demonstrated on MS COCO. These evaluations underscore the method's versatility and applicability in both labeled and unlabeled contexts.

Regarding computational efficiency, we designed our approach to avoid the high costs of fine-tuning by leveraging a lightweight neural prior and a token-sized learnable patch. Compared to methods like LoRA, our method involves far fewer parameters while maintaining the frozen state of the ViT. The learned marker not only achieves computational efficiency but also enhances interpretability by revealing how ViTs’ attention mechanisms can be manipulated to focus on specific locations.

Finally, while suggestions such as testing multiple markers or exploring further ablations are valuable, they are outside the primary scope of our work. Our focus is on learning a single optimal marker to redirect attention effectively. The inclusion of results with the "red circle" as a baseline, along with detailed ablations on marker size, shape, and placement, demonstrates the robustness and effectiveness of our approach. Comparisons with cropping baselines and performance improvements across diverse models further validate the generalizability of our method.

We hope these clarifications address the concerns raised and highlight the depth and innovation of our contributions. We thank the reviewers again for their feedback and look forward to incorporating these suggestions in future work.

---

### Meta-Review · Area_Chair_PRt1 · 2024-12-17

**Metareview:**

The paper received four expert reviews with 3x reject and 1x borderline reject ratings. Overall, the reviewers are concerned about limited technical novelty, limited effectiveness, limited application scenarios of the method, and unjustified results. The authors are suggested to rethink the formulation of the problem domain and improve the manuscript based on the reviewers' comments.

**Additional Comments On Reviewer Discussion:**

The rebuttal fails to provide concrete evidence to answer the reviewers' questions.

---

### Decision · Program_Chairs · 2025-01-22

Reject